# Use of artificial intelligence in obstetric and gynaecological diagnostics: a protocol for a systematic review and meta-analysis

Anjalee Chaurasia [ORCID],[1] Georgia Curry [ORCID],[1] Yi Zhao,[1] Fatema Dawoodbhoy,[2] Jennifer Green,[3] Matilde Vaninetti,[4] Nishel Shah [ORCID],[5,6] Orene Greer[5,6]

¹School of Medicine, Imperial College London, London, UK
²Barking Havering and Redbridge Hospitals NHS Trust, Romford, UK
³Department of Obstetrics & Gynaecology, North West Anglia NHS Foundation Trust, Peterborough, UK
⁴Cambridge University Hospitals NHS Foundation Trust, Cambridge, UK
⁵Department of Metabolism, Digestion and Reproduction, Chelsea and Westminster Hospital, London, UK
⁶Chelsea and Westminster Hospital NHS Foundation Trust, London, UK

**Correspondence to**
Anjalee Chaurasia;
anjalee.chaurasia18@gmail.com

## ABSTRACT

**Introduction** Emerging developments in applications of artificial intelligence (AI) in healthcare offer the opportunity to improve diagnostic capabilities in obstetrics and gynaecology (O&G), ensuring early detection of pathology, optimal management and improving survival. Consensus on a robust AI healthcare framework is crucial for standardising protocols that promote data privacy and transparency, minimise bias, and ensure patient safety. Here, we describe the study protocol for a systematic review and meta-analysis to evaluate current applications of AI in O&G diagnostics with consideration of reporting standards used and their ethical implications. This protocol is written following the Preferred Reporting Items for Systematic Review and Meta-Analysis Protocols (PRISMA-P) 2015 checklist.

**Methods and analysis** The study objective is to explore the current application of AI in O&G diagnostics and assess the reporting standards used in these studies. Electronic bibliographic databases MEDLINE, EMBASE and Cochrane will be searched. Study selection, data extraction and subsequent narrative synthesis and meta-analyses will be carried out following the PRISMA-P guidelines. Included papers will be English-language full-text articles from May 2015 to March 2024, which provide original data, as AI has been redefined in recent literature. Papers must use AI as the predictive method, focusing on improving O&G diagnostic outcomes.

We will evaluate the reporting standards including the risk of bias, lack of transparency and consider the ethical implications and potential harm to patients. Outcome measures will involve assessing the included studies against gold-standard criteria for robustness of model development (Transparent Reporting of a multivariable prediction model for Individual Prognosis Or Diagnosis, model predictive performance, model risk of bias and applicability (Prediction model Risk Of Bias Assessment Tool and study reporting (Consolidated Standards of Reporting Trials-AI) guidance.

**Ethics and dissemination** Ethical approval is not required for this systematic review. Findings will be shared through peer-reviewed publications. There will be no patient or public involvement in this study.

**PROSPERO registration number** CRD42022357024 .

## STRENGTHS AND LIMITATIONS OF THIS STUDY

⇒ To our knowledge, this will be the first comprehensive systematic review and meta-analysis of the ethical considerations of the application of artificial intelligence in obstetrics and gynaecology diagnostics.

⇒ Statistical analysis will be performed to determine the effect of heterogeneity in the literature.

⇒ Heterogeneity of study methods may impact the strength of the conclusions.

⇒ The analysis may be limited if a small study number meets the inclusion criteria.

## BACKGROUND

Access to accurate diagnostics is fundamental to reducing morbidity and mortality. However, common diagnostic tools used in areas of obstetrics and gynaecology (O&G) have their limitations. Many of the diagnostics are performed using ultrasound, which is subjective and prone to error.[1 2] For instance, interoperator and intraoperator variation in fetal birth weight estimates and discrepancies in nuchal translucency are acknowledged.[3–5] Another example is in the tools used to predict preterm birth, a major cause of neonatal death worldwide. Additional work is required to improve our ability to accurately predict it and subsequently improve clinical outcomes[6 7]; particularly as much of the resultant mortality might be prevented with cost-effective interventions.[8–10] Thus, emerging technological artificial intelligence (AI) developments in medicine may offer an opportunity to improve diagnostic capabilities in O&G.[11]

AI uses multifactorial data to build connections and learn. Such data can take the form of images or numeric values obtained from existing databases or laboratory tests.[12] For example, machine learning models can be applied to a series of medical images to extract

key features associated with specific pathologies and this can be used to increase the accuracy of detecting gynaecological malignancy, for example, malignant epithelial ovarian tumours.[13] It has been suggested that AI has the potential to perform problem-solving at a greater efficiency than humans.[14]

However, the development in AI technology requires parallel development of rigorous protocols, which can be applied across AI applications, to ensure data privacy, transparency, safety and fairness, as well as to minimise bias. The importance of improved governance of AI research can be demonstrated by the example of an algorithm, which was developed to determine patient health risks for approximately 200 million individuals in the USA each year.[15] It was shown to exhibit racial bias, with the choice of label used to train this data resulting in fewer black patients receiving the care they needed.[15]

The implementation of this new technology, if unregulated, has the potential to cause personal and societal harm, with risk to patient safety.[16] Foremost, AI algorithms may be subject to predictive error due to the training data set used. This may result in incorrect clinical findings and subsequent undertreatment or overtreatment, unknown to both clinician and patient.[17] Therefore, it is important that researchers developing AI models acknowledge the ethical implications AI may pose for healthcare and adhere to recommended standards for reporting their methods and data. We aim to evaluate current practice and highlight where work is required to improve this. Thus, we will systematically review existing studies, assessing reporting standards, performance, model development and performance, and adherence to 'gold-standard' criteria.

## METHODS AND ANALYSIS

The protocol is registered with the international Prospective Register of Systematic Reviews (PROSPERO), registration number CRD42022357024. The protocol and the completed systematic review will be reported using the Preferred Reporting Items for Systematic Reviews and Meta-analyses for Protocols (PRISMA).[18]

### Search methodology

The search will be carried out using Medical Literature Analysis and Retrieval System Online (MEDLINE), Excerpta Medica Database (EMBASE) and Cochrane. The search strategy will include Medical Subject Headings (MESH) terms, free text and Boolean operators. We will search using the terms "obstetrics" and "gynaecology", multiple synonyms of "artificial intelligence" including "machine learning", "deep learning" and "algorithm", along with "diagnostics", "prognostics" and their variations. The search strategy will be adapted to the specific database. A full-search strategy can be found in online supplemental file. A manual search of the grey literature will also be performed. The studies included will only be in the English language, published from May 2015 to March 2024, as AI methods were recently redefined in a seminal article published in 2015.[19]

### Study selection and data extraction

The studies will be uploaded to a systematic review study management software, for example, Rayyan,[20] to remove duplicates. The studies will be screened by two independent reviewers, comparing the titles and abstracts to the inclusion criteria. The same process will then be carried out for the full text of the selected studies. Any disagreements between the reviewers will be resolved via discussion or following discussion with an additional reviewer. Study exclusion criteria will be recorded and reported in the review. The references of manuscripts identified from the search will also be screened for studies suitable for inclusion. A PRISMA flow chart of the search and study selection will be included in the report.[21]

### Inclusion criteria

This systematic review will include peer-reviewed studies that use patient data for predictive outcomes. To be included in the analysis, studies must use AI as the predictive method, focusing on improving O&G diagnostic outcomes and be written in English.

### Exclusion criteria

Non-human studies and papers that do not use patient data for predictive outcomes. Review articles, commentaries, abstracts, meta-analyses, case reports and expert opinions will be excluded. Literature that does not discuss AI in O&G diagnostics and papers not published in the English language will also be excluded.

### Data extraction

Data will be extracted from the studies using Excel. The data extracted will include the year of publication, study authors, study design, patient population, study size, the O&G pathology evaluated, the diagnostic technique, type of AI model, reporting system for example Transparent Reporting of a multivariable prediction model for Individual Prognosis Or Diagnosis (TRIPOD)-AI, the performance measurements used, the method of clinical deployment of the AI model and whether ethical approval was granted for each study. In cases of incomplete or missing data in published studies, the authors will be contacted to capture the full data for extraction. Table 1 summarises the data to be extracted.

### Endpoints

The primary endpoint is a demonstration of the application of current recommended standards for the development and reporting of AI primary research involving O&G diagnostics (relative to Consolidated Standards of Reporting Trials-AI, TRIPOD and Prediction model Risk Of Bias Assessment Tool (PROBAST) guidelines).[22] We will summarise these findings in a table.

**Table 1** Data collection items

| Item no. | Data title | Data type |
|---|---|---|
| 1 | Year of publication | Study characteristic |
| 2 | Study authors | Study characteristic |
| 3 | Study design | Study characteristic |
| 4 | Patient population | Demographics |
| 5 | Study size | Demographics |
| 6 | Target O&G pathology | Methodology |
| 7 | Diagnostic technique | Methodology |
| 8 | Type of AI model | Outcome |
| 9 | Reporting system used | Outcome |
| 10 | Method of illustrating diagnostic performance | Outcome |
| 11 | Method of model deployment | Outcome |
| 12 | Whether ethical approval was granted for each study (Y/N) | Outcome |

AI, artificial intelligence; O&G, obstetrics and gynaecology.

## Meta-analysis

If there are sufficient papers illustrating the diagnostic performance of the AI models for diseases relating to the field of O&G, a meta-analysis will be conducted. Model performance measures, including sensitivity, specificity and the area under (AUC) the receiver operating curve values for specific O&G endpoints will be compared in the meta-analysis. The pooled sensitivity and specificity can be compared with the reference standards and reported. If any of the required data has not been reported, we will contact the study authors to request data sharing. If studies present a different metric for model performance, these will also be included in the report.

A 2×2 contingency table containing true positive, false positive, true negative, false negative will be created for the sensitivity and specificity of each study and compared using bivariate analysis. These will subsequently be plotted on forest plots and summary receiver operating characteristic (sROC) curve plots. The sROC will be calculated with the individual AUC.

Interstudy variation and heterogeneity will be displayed using $I^2$. To obtain pooled estimates, we will use a random-effects model. Leave-one-out analyses will be used to detect outliers, and the model will be changed accordingly. To control for study variance, we will use untransformed, logit and double-arcsine transformed proportions, and normality will be assessed using the Shapiro-Wilk test and density plots. Ratios resembling normal distributions will be used. Analysis and visualisation will be carried out using R statistical environment (V.3.6.1, 2019-07-05).

## Risk of bias in individual studies

PROBAST will be used to assess the included studies for bias and applicability.[22] It is divided into four domains:

participants, predictors, outcome and analysis. It has 20 questions that help to judge bias. Bias occurs when the shortcomings in the study design, conduct or analysis impact predictive performance. It will give an idea of the reliability of the data produced and the appropriateness of the model for diagnosis and prognosis. If studies with high bias are included, the impact and reasons for this will be discussed.

## Patient and public involvement

There will be no patient or public involvement in this study.

## Trial status

- ► Preliminary searches: started .
- ► Piloting of the study selection process: started .
- ► Formal screening: not started .
- ► Data extraction: not started .
- ► Risk of bias assessment: not started .
- ► Data analysis: not started .

## ETHICS AND DISSEMINATION

There are no ethical concerns as the review does not require consent or subject participation. Dissemination of the protocol and systematic review will be done through a peer-reviewed journal.

**Acknowledgements** We would like to acknowledge and thank Mr Phillip Barlow who assisted with optimising and refining our proposed search strategy. Citation: Evidence search: Machine learning for use in diagnostic tools for obstetrics and gynaecology. Phillip Barlow (Imperial College Library Services) (6 March 2024) LONDON, UK: Imperial College NHS and Chelsea & Westminster Hospital NHS Trusts Libraries.

**Contributors** The authors' contribution includes but is not limited to, the following: GC, AC and YZ drafted the manuscript and created the study concept. OG, NS, FD, JG and MV provided supervision and guidance during the study. All authors reviewed and approved the manuscript in its current form. GC and AC are the guarantors of this work.

**Funding** The study was funded by Imperial College London Open Access Fund.

**Competing interests** None declared.

**ORCID iDs**

Anjalee Chaurasia http://orcid.org/0000-0003-3924-9201

Georgia Curry http://orcid.org/0000-0002-2888-3859
Nishel Shah http://orcid.org/0000-0002-5694-9915

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
