## [Reviewer comments · BMJ Open]

ARTICLE DETAILS

TITLE (PROVISIONAL)	The use of artificial intelligence in obstetric and gynaecological diagnostics: a protocol for a systematic review and meta-analysis
AUTHORS	Chaurasia, Anjalee; Curry, Georgia; Zhao, Yi; Dawoodbhoy, Fatema; Green, Jennifer; Vaninetti, Matilde; Shah, Nishel; Greer, Orene

VERSION 1 – REVIEW

REVIEWER	Matsuoka, Ryu Showa University Graduate School of Medicine School of Medicine
REVIEW RETURNED	22-Dec-2023

GENERAL COMMENTS	I would like an explanation for using the 2015 version of the PRISMA checklist in this paper, even though the 2020 version has already been published.
--

REVIEWER	Alcázar, Juan Clínica Universidad de Navarra
REVIEW RETURNED	02-Jan-2024

GENERAL COMMENTS	Interesting and well written manuscript. I have minor comments 1. There are some redundant sentences. For example, the objective is stated twice in the Abstract. 2. The authors claim this is the first meta-analysis about the use of AI in Ob/Gyn. This is not true. There are other meta-analyses for some specific areas in the field of Ob/Gyn 3. Why limiting language in the search? 4. Were the authors contacted in case of missing information?
---

VERSION 1 – AUTHOR RESPONSE

Please find below our responses to the editor/ reviewer comments:

1. Please clarify in the methods section why the search is restricted to May 2015 onwards. Thank you for your comment. We have restricted our search to May 2015 as we consider that this point marks a milestone in the literature for clarification of an updated definition of artificial intelligence (AI), machine learning and convolutional neural networks, in terms of the computing methods used for subsequent AI projects. We consider that, by using this

timepoint, we will increase the likelihood of an evaluation of studies with methodologies which can be comparatively evaluated. We have clarified this in line 41 of the manuscript.

2. Please include, as a supplementary file, a draft of the precise search strategy (or strategies) for all databases, registers and websites, including any filters and limits used.
Thank you, we have updated our draft search strategy to demonstrate greater precision for the databases we intend to use and this has been removed from the main paper to a separate supplementary file.
3. I would like an explanation for using the 2015 version of the PRISMA checklist in this paper, even though the 2020 version has already been published.
Thank you kindly for your comment Dr Matsuoka. The 2015 version is the PRSIMA-protocol (PRISMA-P) checklist not the full PRISMA check list <https://www.bmj.com/content/349/bmj.g7647>. We are using the protocol checklist specifically for our protocol of the systematic review. However, when constructing the systematic review we will be using the most up-to-date PRISMA.
4. There are some redundant sentences. For example, the objective is stated twice in the Abstract.
Thank you kindly, we have taken out the repeated sentence in line 32-34.
5. The authors claim this is the first meta-analysis about the use of AI in Ob/Gyn. This is not true. There are other meta-analyses for some specific areas in the field of Ob/Gyn
Thank you kindly for your clarification. We wish to state that we will place an emphasis on the ethical considerations in conjunction with our meta-analysis results, as stated in Line 56-57. We believe that if a tool using model prediction is used to guide patients' management, thus impacting clinical outcomes, the ethics surroundings the development, deployment and regulations should be thoroughly reviewed. We believe that this is the first meta-analysis which will discuss the ethical implications of using these tools in the field of obstetrics & gynaecology. We have indicated this on line 56 of the manuscript.
6. Why limiting language in the search?
Unfortunately, this is a limitation of our systematic review, as all authors will be English speakers. We do acknowledge that this will result in studies published in languages other than English being omitted from our full search and acknowledge this as a limitation on the body of literature that we will be able to fully review.
7. Were the authors contacted in case of missing information?
Thank you for highlighting the need to address potential missing data for this project. We intend to contact study authors to request data sharing to minimise this and have highlighted this in line 146-147 of the manuscript.

The updates have been registered by "track changes" and we will also submit an updated version of the manuscript.